# Assessing the Impact of Vaccination on the Dynamics of COVID-19 in Africa: A Mathematical Modeling Study

**DOI:** 10.3390/vaccines11040857

**Published:** 2023-04-17

**Authors:** Yvette Montcho, Robinah Nalwanga, Paustella Azokpota, Jonas Têlé Doumatè, Bruno Enagnon Lokonon, Valère Kolawole Salako, Martin Wolkewitz, Romain Glèlè Kakaï

**Affiliations:** 1Laboratoire de Biomathématiques et d’Estimations Forestières, Université d’Abomey-Calavi, Cotonou 04 BP 1525, Benin; 2Faculté des Sciences et Techniques, Université d’Abomey-Calavi, Abomey-Calavi, Cotonou 01 BP 526, Benin; 3Institute of Medical Biometry and Statistics, Faculty of Medicine and Medical Center, University of Freiburg, 79104 Freiburg, Germany

**Keywords:** COVID-19, vaccination impact, compartmental model, reproduction number, Africa

## Abstract

Several effective COVID-19 vaccines are administered to combat the COVID-19 pandemic globally. In most African countries, there is a comparatively limited deployment of vaccination programs. In this work, we develop a mathematical compartmental model to assess the impact of vaccination programs on curtailing the burden of COVID-19 in eight African countries considering SARS-CoV-2 cumulative case data for each country for the third wave of the COVID-19 pandemic. The model stratifies the total population into two subgroups based on individual vaccination status. We use the detection and death rates ratios between vaccinated and unvaccinated individuals to quantify the vaccine’s effectiveness in reducing new COVID-19 infections and death, respectively. Additionally, we perform a numerical sensitivity analysis to assess the combined impact of vaccination and reduction in the SARS-CoV-2 transmission due to control measures on the control reproduction number ($R_{c}$). Our results reveal that on average, at least 60% of the population in each considered African country should be vaccinated to curtail the pandemic (lower the $R_{c}$ below one). Moreover, lower values of $R_{c}$ are possible even when there is a low (10%) or moderate (30%) reduction in the SARS-CoV-2 transmission rate due to NPIs. Combining vaccination programs with various levels of reduction in the transmission rate due to NPI aids in curtailing the pandemic. Additionally, this study shows that vaccination significantly reduces the severity of the disease and death rates despite low efficacy against COVID-19 infections. The African governments need to design vaccination strategies that increase vaccine uptake, such as an incentive-based approach.

## 1. Introduction

Coronavirus disease 2019 (COVID-19) caused by severe acute respiratory syndrome coronavirus-2 (SARS-CoV-2) [1] invaded the world unexpectedly in 2019 and changed human life tremendously [2]. The disease outbreak first emerged in Wuhan City, Hubei Province of China [3], and after that, it spread to the United States, Europe, Asia, and later on, to other continents. Despite its rampant spread, studies indicated that the spread of the disease in Africa had not followed an exponential path as for the rest of the world (Europe, United States, Asia), implying that Africa has not yet experienced the predicted heavy disease burden [4]. By the end of November 2022, approximately 12.7 million cases and 257,984 deaths, representing 2.1% and 4.3%, respectively of the global figures, were reported in Africa [5]. In an effort to curb the transmission of SARS-CoV-2, many African countries have implemented non-pharmaceutical interventions (NPIs), such as social distancing, quarantine of suspected infection cases, use of face masks, contact tracing and testing, among others [6]. Several studies [7,8,9] have investigated the effectiveness of NPIs on the transmission dynamics of COVID-19 using various approaches. Findings from these studies have indicated that NPIs have been sufficiently effective in mitigating the burden of the pandemic, at least for the first and second waves. However, the emergence of new SARS-CoV-2 variants, which are currently categorized as Variants of Concern (VOC) by the WHO, such as Alpha, Beta, Omicron, and Delta strains [10] have reduced the effectiveness of NPIs, creating the need for more effective control measures such as vaccination [11].

Vaccination against COVID-19 has been identified as one of the most viable options to suppress the SARS-CoV-2 transmission globally [12] and as well achieve herd immunity [13]. Several COVID-19 vaccines have been approved for use, and they are commonly administered as either a single dose such as Johnson & Johnson (52.0–72.0% of efficacy) or two-doses vaccine such as AstraZeneca (62.1–90.0%), Pfizer-BioNTech (95.0%) and Moderna (94.1%) [14,15]. The first mass vaccination program began in early 2020. By the end of November 2022, more than 5.46 billion vaccine doses have been administered worldwide, representing 71.1% of the global population [16]. Over 2.6 million additional doses (boosters) have been administered to fully vaccinated people [16]. COVID-19 vaccination rates remain low in most African countries. As of November 2022, only 33% of the population had received at least one dose of the vaccine compared to the global average of 69.0% [16]. Vaccine hesitancy due to widespread misconceptions and beliefs about vaccines, a lack of adequate infrastructure and logistics to handle vaccination campaigns, and a low-risk perception of the pandemic, notably with the recent decline in cases, are the major barriers to low vaccination uptake in Africa [17,18]. Several clinical trials have shown that COVID-19 vaccines are effective in reducing disease severity and individual symptoms, decreasing fatalities, hospitalizations, and admissions to intensive care units [19,20]. However, the emergence of new variants may outweigh some of these gains. Given the contagious nature of infectious diseases, particularly COVID-19, there is mounting evidence that poor vaccine uptake may not only amplify disease transmission in unvaccinated subpopulations but also heighten the risk for vaccinated populations, especially in situations where vaccines confer imperfect immunity. A recent study by the US Centers for Disease Control and Prevention on an outbreak of COVID-19 in a federal prison in Texas showed an equal transmission rate among vaccinated and unvaccinated individuals [21]. Mathematical models are important tools to describe and predict the spread of epidemics and can be used to quantify the potential impact of vaccination programs on disease dynamics. Currently, several mathematical models have been developed to predict and assess the impact of vaccination on the transmission dynamics of COVID-19 [22,23,24,25,26].

Machado et al. [27] analyzed the impact of vaccination on the control of the pandemic using a simple SEIR-based simulation model. The authors believe that an increased vaccination rate combined with continued adherence to non-pharmaceutical interventions can greatly delay the peak of infection. With the ongoing vaccination program, the trajectory of a pandemic is determined by how the virus spreads in unvaccinated and vaccinated individuals. The effect of mixing vaccinated and unvaccinated populations on the risk of SARS-CoV-2 infection among vaccinated people was investigated in a study by [28] using a basic SIR model. Under all mixing assumptions, their model demonstrated that the risk of infection was significantly higher in the unvaccinated group than in the vaccinated group. However, the author acknowledges that the simplicity of the model does not reflect the real-world process of the pandemic, for instance, demographics, waning of the vaccine, and natural loss of immunity, among others.

Even though various research studies are being conducted in Africa, they largely focus on COVID-19 vaccination prioritization strategies [29,30,31] or vaccine acceptance [32,33,34]. To the best of our knowledge, however, studies that use real-world vaccination data to evaluate the impact of COVID-19 vaccination programs on the dynamics of the disease in African countries are still very few [35]. Despite the contributions of these studies, they showed some limitations. Thus, the previous studies ignore the fact that transmission can occur both within vaccinated and unvaccinated people and between vaccinated and unvaccinated people (i.e., an infectious unvaccinated person can infect a vaccinated person and vice versa).

In this study, we developed a mathematical model to quantitatively assess the impact of vaccination programs on the dynamics of COVID-19 in Africa, focusing on eight countries (Algeria, DR Congo, Kenya, Lybia, Namibia, Nigeria, Rwanda, and South Africa), representing the four main regions of the continent. The model was used to (i) assess the impact of vaccination on COVID-19 incidence and mortality in a mixed population of vaccinated and unvaccinated individuals and (ii) evaluate the combined impact of vaccination with different levels of NPIs on the dynamics of COVID-19.

## 2. Materials and Methods

### 2.1. Model Formulation

In this study, we developed a deterministic compartmental model of COVID-19 stratified by infection status and vaccination status to describe the impact of an imperfect vaccine on the transmission dynamics of the disease. The proposed model is a modification of a previously developed compartmental model [36] where vaccination was incorporated in the COVID-19 model as a pharmaceutical intervention strategy in South Africa. The mathematical model comprises eight epidemiological states depending on the individual’s health and vaccination status. The total population at time t denoted by N(t) is divided into two groups, i.e., unvaccinated and vaccinated, which are represented by subscripts *u* and *v*, respectively.

The unvaccinated population denoted by N(u) is further subdivided into eight subpopulations of individuals that are: unvaccinated susceptible ($S_{u}\left(t\right)$), unvaccinated exposed ($E_{u}\left(t\right)$), unvaccinated pre-symptomatic infectious ($I_{p_{u}}\left(t\right))$, unvaccinated asymptomatic infectious ($I_{A_{u}}\left(t\right)$), unvaccinated symptomatic infectious ($I_{S_{u}}\left(t\right)$), the detected infectious unvaccinated via testing ($C_{u}\left(t\right)$), unvaccinated recovered ($R_{u}\left(t\right)$), and unvaccinated deceased $D_{u}\left(t\right)$. Thus, the total population for the unvaccinated is given by:$$N_{u}\left(t\right)=S_{u}\left(t\right)+E_{u}\left(t\right)+I_{p_{u}}\left(t\right)+I_{A_{u}}\left(t\right)+I_{S_{u}}\left(t\right)+C_{u}\left(t\right)+R_{u}\left(t\right).$$

Similarly, the vaccinated population denoted by N(v) is also further subdivided into eight subpopulations of individuals that are: vaccinated susceptible ($S_{v}\left(t\right)$), vaccinated exposed ($E_{v}\left(t\right)$), vaccinated pre-symptomatic infectious ($I_{p_{v}}\left(t\right))$, vaccinated asymptomatic infectious ($I_{A_{v}}\left(t\right)$), vaccinated symptomatic infectious ($I_{S_{v}}\left(t\right)$), detected infectious vaccinated via testing ($C_{v}\left(t\right)$), vaccinated recovered ($R_{v}\left(t\right)$) and vaccinated deceased $D_{v}\left(t\right)$. The total population for the vaccinated is given by:$$N_{v}\left(t\right)=S_{v}\left(t\right)+E_{v}\left(t\right)+I_{p_{v}}\left(t\right)+I_{A_{v}}\left(t\right)+I_{S_{v}}\left(t\right)+C_{v}\left(t\right)+R_{v}\left(t\right).$$

Therefore, the total population at time t (denoted by N(t)) is
$$N\left(t\right)=N_{u}\left(t\right)+N_{v}\left(t\right).$$

When developing the mathematical model, we made some assumptions or comments, which are as follows.

(i)Vaccination is administered to unvaccinated individuals that are susceptible, exposed, pre-symptomatic, asymptomatic, and naturally recovered from the virus. The model does not consider the vaccination of symptomatic and confirmed infectious individuals.(ii)The COVID-19 vaccine administered is imperfect, i.e., it provides only partial protection against COVID-19 infections. Thus, infections for the vaccinated can occur but at a reduced rate compared to that of the unvaccinated susceptible individuals.(iii)Both vaccine-derived and natural immunity may wane over time in individuals, implying that individuals rejoin the fully susceptible class after a certain period [36,37,38].(iv)We assume that there is homogeneous mixing among the population, which means that every individual in the community is equally likely to mix and acquire infections from each member when they make contact.(v)Since the COVID-19 pandemic has persisted for a long time, we include the vital dynamics (birth and natural death) in the model.

We suppose that all the births and the immigration from the population are recruited into the unvaccinated susceptible class at rate Λ. Susceptible unvaccinated individuals become exposed following effective contact with either unvaccinated or vaccinated pre-asymptomatic, asymptomatic, symptomatic, and confirmed infectious individuals at a rate $\lambda_{u}$. After the latent period, the unvaccinated exposed individuals become pre-asymptomatic at a progression rate $\alpha_{E}$. At the end of the incubation period, unvaccinated pre-symptomatic infectious individuals either develop clinical symptoms and move to the unvaccinated symptomatic infectious ($I_{S_{u}}$) at a rate $\rho_{1}\alpha_{p}$ (where $\rho_{1}$ is the probability of developing symptoms), or they continue to show no symptoms and move on to the unvaccinated asymptomatic class ( $I_{A_{u}}$) at the rate $\left(1-\rho_{1}\right)\alpha_{p}$. The asymptomatic and symptomatic unvaccinated infectious individuals are tested and confirmed positive at a detection rate $q_{a_{1}}$ and $q_{s_{1}}$, respectively, and move to the unvaccinated confirmed class ($C_{u}$). The symptomatic and confirmed unvaccinated infectious individuals might die due to COVID-19-related complications at the rate $\delta_{s_{1}}$ and $\delta_{c_{1}}$, respectively. The parameters $\gamma_{a_{1}}$, $\gamma_{s_{1}}$ and $\gamma_{c_{1}}$ account for the recovery rates for unvaccinated individuals in the asymptomatic, symptomatic and confirmed classes, respectively. The recovered unvaccinated individuals may lose their natural immunity at a rate $d_{u}$, and thus, they can become susceptible.

We assume that unvaccinated individuals in the susceptible, exposed, pre-symptomatic, asymptomatic and recovered classes are vaccinated at rate ν. Due to the imperfect vaccine administrated, vaccinated individuals are not immune from infection. The vaccine-induced immunity of the susceptible vaccinated individuals wanes at a per capita rate ω. Hence, after a given time, the susceptible vaccinated population can become infected by the virus when they make contact with either unvaccinated or vaccinated pre-asymptomatic, asymptomatic, symptomatic, and confirmed infectious individuals at a rate $\lambda_{v}$. The population in the class $E_{v}$ becomes infectious at a rate $\alpha_{E}$ and moves to the pre-asymptomatic class $I_{p_{v}}$. After the pre-asymptomatic period, proportion $\rho_{2}$ develops COVID-19 symptoms and moves to the symptomatic infectious class ($I_{S_{v}}$), while the rest continue to show no symptoms and move on to the vaccinated asymptomatic class ($I_{A_{v}}$). The asymptomatic and symptomatic vaccinated infectious individuals are tested and confirmed positive at a detection rate $q_{a_{2}}$ and $q_{s_{2}}$, respectively, and move to the vaccinated confirmed class ($C_{v}$). The symptomatic and confirmed vaccinated infectious individuals may die due to COVID-19-related complications at the rate $\delta_{s_{2}}$ and $\delta_{c_{2}}$. The parameters $\gamma_{a_{2}}$, $\gamma_{s_{2}}$ and $\gamma_{c_{2}}$ account for the recovery rates of vaccinated individuals in the asymptomatic, symptomatic and confirmed classes. The recovered vaccinated individuals may lose derived vaccine immunity at a rate $d_{v}$, and thus, they can become susceptible. Each subpopulation is reduced by a natural death at a constant rate μ.

The flowchart of the formulated model using all the above assumptions is given in Figure 1. Additionally, all the model state variables and the parameters with their description are presented in Table 1 and Table 2, respectively. Hence, the COVID-19 dynamics for the unvaccinated population are described by the following system of differential equations:(1)$$\begin{array}{cc} {\dot{S}}_{u} & =\Lambda -\left(\lambda_{u}+\mu +\nu \right)S_{u}+S_{v}\omega +d_{u}R_{u},\\ {\dot{E}}_{u} & =\lambda_{u}S_{u}-\left(\alpha_{E}+\nu +\mu \right)E_{u},\\ {\dot{I}}_{P_{u}} & =\alpha_{E}E_{u}-\left(\alpha_{p}+\mu +\nu +q_{p_{1}}\right)I_{P_{u}},\\ {\dot{I}}_{A_{u}} & =\left(1-\rho_{1}\right)\alpha_{p}I_{P_{u}}-\left(\mu +\nu +\gamma_{a_{1}}+q_{a_{1}}\right)I_{A_{u}},\\ {\dot{I}}_{S_{u}} & =\rho_{1}\alpha_{p}I_{P_{u}}-\left(\mu +\gamma_{s_{1}}+q_{s_{1}}+\delta_{s_{1}}\right)I_{S_{u}},\\ {\dot{C}}_{u} & =q_{a_{1}}I_{A_{u}}+q_{s_{1}}I_{S_{u}}+q_{p_{1}}I_{P_{u}}-\left(\delta_{c_{1}}+\gamma_{c_{1}}+\mu \right)C_{u},\\ {\dot{D}}_{u} & =\delta_{s_{1}}I_{S_{u}}+\delta_{c_{1}}C_{u},\\ {\dot{R}}_{u} & =\gamma_{a_{1}}I_{A_{u}}+\gamma_{s_{1}}I_{S_{u}}+\gamma_{c_{1}}C_{u}-\left(d_{u}+\mu +\nu \right)R_{u}, \end{array}$$
where $\lambda_{u}$ is the force of infection for the unvaccinated individuals, which is defined by:$$\begin{array}{cc} \lambda_{u}= & \frac{b_{uu}\left(\theta_{P_{u}}I_{P_{u}}+\theta_{A_{u}}I_{A_{u}}+\theta_{s_{u}}I_{S_{u}}+\theta_{c_{u}}C_{u}\right)\left(1-\psi_{u}\right)}{N},\\ & +\frac{b_{uv}\left(\theta_{P_{v}}I_{P_{v}}+\theta_{A_{v}}I_{A_{v}}+\theta_{S_{v}}I_{S_{v}}+\theta_{c_{v}}C_{v}\right)\left(1-\psi_{v}\right)}{N}. \end{array}$$

Similarly, using the same model assumptions and parameter description, the COVID-19 dynamics for the vaccinated population are described by the following system of differential equations:(2)$$\begin{array}{cc} {\dot{S}}_{v} & =\nu S_{u}+d_{v}R_{v}-\left(\lambda_{v}+\mu +\omega \right)S_{v},\\ {\dot{E}}_{v} & =\nu E_{u}+\lambda_{v}S_{v}-\left(\alpha_{E}+\mu \right)E_{v},\\ {\dot{I}}_{P_{v}} & =\nu I_{P_{u}}+\alpha_{E}E_{v}-\left(\alpha_{p}+q_{p_{2}}+\mu \right)I_{P_{v}},\\ {\dot{I}}_{A_{v}} & =\nu I_{A_{u}}+\left(1-\rho_{2}\right)\alpha_{p}I_{P_{v}}-\left(\mu +\gamma_{a_{2}}+q_{a_{2}}\right)I_{a_{v}},\\ {\dot{I}}_{S_{v}} & =\rho_{2}\alpha_{p}I_{P_{v}}-\left(\mu +\gamma_{s_{2}}+q_{s_{2}}+\delta_{s_{2}}\right)I_{S_{v}},\\ {\dot{C}}_{v} & =q_{a_{2}}I_{A_{v}}+q_{s_{2}}I_{S_{v}}+q_{p_{2}}I_{P_{v}}-\left(\delta_{c_{2}}+\gamma_{c_{2}}+\mu \right)C_{v},\\ {\dot{D}}_{v} & =\delta_{s_{2}}I_{S_{v}}+\delta_{c_{2}}C_{v},\\ {\dot{R}}_{v} & =\nu R_{u}+\gamma_{a_{2}}I_{A_{v}}+\gamma_{s_{2}}I_{S_{v}}+\gamma_{c_{2}}C_{v}-\left(d_{v}+\mu \right)R_{v}, \end{array}$$
where $\lambda_{v}$ is the force of infection for the vaccinated individuals, which is defined by;
$$\begin{array}{cc} \lambda_{v} & =\frac{b_{vu}\left(\theta_{P_{u}}I_{P_{u}}+\theta_{A_{u}}I_{A_{u}}+\theta_{S_{u}}I_{S_{u}}+\theta_{c_{u}}C_{u}\right)\left(1-\psi_{u}\right)}{N},\\ & +\frac{b_{vv}\left(\theta_{P_{v}}I_{P_{v}}+\theta_{A_{v}}I_{A_{v}}+\theta_{S_{v}}I_{S_{v}}+\theta_{c_{v}}C_{v}\right)\left(1-\psi_{v}\right)}{N}, \end{array}$$
where $0<\psi_{u}<1$ and $0<\psi_{v}<1$ represent the percentage decrease in the transmission rate due to control measures among the unvaccinated and vaccinated individuals, respectively.

### 2.2. Data

Five countries per African region were randomly selected among those for which COVID-19 data are available. However, during the modeling process, two countries, namely Benin and Gabon, were excluded due to the poor quality of the data. Consequently, eight African countries, namely, DR Congo, Rwanda, Kenya, Algeria, Libya, Namibia, South Africa, and Nigeria, were selected for analysis in this study. Data on the daily COVID-19 cases, cumulative confirmed cases, and vaccination (number of individuals vaccinated with at least one dose) for each selected country were obtained from COVID-19 data respiratory by Our World in Data (https://github.com/owid/covid-19-data/tree/master/public/data, accessed on 15 july 2022).

The country-specific demographic data such as birth rates and death rates were obtained from the Worldbank via (https://data.worldbank.org/indicator/SP.DYN.CBRT.IN, accessed on 15 july 2022), while data on annual net migration and life expectancy for each country were obtained from Worldmeter, which was available via (https://www.worldometers.info/world-population/population-by-country/, accessed on 15 july 2022).

### 2.3. Model Fitting and Parameter Estimation Procedure

In this subsection, a single model with sixteen compartments was used for the calibration. We consider a mixed population where both vaccinated and unvaccinated individuals interact, and the transition from the unvaccinated classes to the vaccinated classes is described by some parameters. Thus, we estimated the best values of unknown parameters in models (Equation 1–Equation 2). We used the data of COVID-19 cumulative cases for each country from the first day of vaccination to the end of the third pandemic wave (end of November 2021). The choice for cumulative case data over daily case data is because it mitigates the effect of reporting errors in modeling COVID-19 dynamics. The start dates and the end dates for each country are presented in Table A1. The fixed parameters used in the model-fitting process were obtained from the literature as presented in Table A2, while other fixed parameters that vary per country were calculated and are presented in Table A3.

We define a vaccinated individual as one who has received at least one dose of the COVID-19 vaccine since the available data only give the new vaccination doses delivered per day and make no distinction between the first and second doses. The vaccination rate, ν, for each country is given by
$$\nu =\frac{\mathrm{Vaccine}\;\mathrm{coverage}}{\mathrm{Vaccination}\;\mathrm{period}},$$
where vaccine coverage is the proportion of individuals vaccinated with at least one dose of the COVID-19 vaccine at the end date for each country.

Two demographic parameters were computed for each country, i.e., daily recruitment rate (Λ) of unvaccinated susceptible (through births and net migrations) per (individuals/day) and the natural death rate (μ) per day were computed.

The daily recruitment rate, Λ for each country, was computed using the following expression [39].
(3)$$\Lambda =\frac{r_{b}\bar{N}}{L},$$
where $r_{b}=\chi_{p}+\frac{A_{I}}{\bar{N}},$ *L* is the vaccination time period for each country, $\chi_{p}$ represents the annual births during the vaccination period, *L* for each country, $\bar{N}$ is the average population size during the vaccination period, *L* in each country and $A_{I}$ represents the net annual migration in the country during vaccination period *L*. Let us take the example of Rwanda. The start date (the first day of vaccination) in Rwanda was 5 March 2021, and the end date is 13 December 2021 (which corresponds to the last day of the third wave of the pandemic). Thus, the vaccination period considered is L=284 days. The mean total population of Rwanda as of 13 December 2021 is $\bar{N}=13,461,888$. Using the annual birth rate (30.725/1000) and the net annual migration (−9000 individuals) in Rwanda, we obtained the rates $r_{b}=\left(30.725/1000\right)-\left(9000/\bar{N}\right)=0.030056$. Using Equation (Equation 3), we computed Λ=1427.7060 individuals/day. For each country, the natural death rate of individuals per day was calculated as the reciprocal of the life expectancy (L.E) at the end date (last day of the third wave of the pandemic). For example, in Rwanda, as of 13 December 2021, the average annual life expectancy was L.E = 70 years, then, the natural death rate was μ=1/(70×365), which gives $\mu =3.9112\times 10^{-5}$ day ${}^{-1}$.

To find the best set of parameters and initial conditions for each country, we used the nonlinear least square technique in Matlab (2021) with *fminsearchbnd,* which is a built-in Matlab function. Here, we minimize the root mean square of squared differences between each observed cumulative case data and the corresponding cumulative case obtained from the model (RMSE1). We repeated this procedure 2000 times to increase the precision of the estimation.

The value for $S_{v0}$ for each country was obtained from the vaccination data, corresponding to the total number of people vaccinated on the first day of vaccination, such that $S_{v0}=N_{v0}$. We suppose that on the first day of vaccination, no individuals are infected with COVID-19 and vaccinated. Then, we set $E_{v}=I_{p_{v}}=I_{A_{v}}=I_{s_{v}}=C_{v}=R_{v}=D_{v}=0$. The solutions to the model Equations (Equation 1) and (Equation 2) were obtained using the built-in function *ODE45* of Matlab. We used the cross-validation technique for parameter estimation to improve the prediction power of the model. To do this, we divided the data into training (90%) and testing (10%) datasets and computed the root mean square error, RMSE1 (computed using the training dataset) and RMSE2 (using the testing dataset), respectively. The whole model was repeated about 100 times, and the final values of the estimates were those with the smallest value for RMSE2 and RMSE1. We also obtained the 95% confidence interval for the estimated parameters considering the normal distribution. The values for initial conditions and the corresponding estimated parameters and their 95% CI are presented in Table A4, Table A5 and Table A6, respectively.

A numerical simulation was carried out to evaluate the impact of vaccination on COVID-19 incidence and mortality in the selected African countries. To quantify the vaccine impact, we determined the vaccine effectiveness in terms of the detection and death rates ratio for vaccinated individuals in relation to unvaccinated, respectively.

Additionally, numerical sensitivity analysis was performed to assess the combined impact of vaccination with different levels of adherence to non-pharmaceutical interventions (NPIs) on the control reproduction number ($R_{c}$). The combined impact was assessed by generating contour plots of control reproduction number ($R_{c}$) as a function of vaccine coverage (VC) and control measures (ψ) among both unvaccinated and vaccinated. We suppose that the implementation and lifting of NPIs are related to changes in the transmission rate. We considered varying levels (0 to 1) of adherence to NIPs by both vaccinated and unvaccinated individuals to represent behaviors that reduce the transmission of the SARS-CoV-2 virus. The level of NPI intensity was categorized as follows: low level (self-protection, use of face masks, hand hygiene, and social distancing), moderate level (mobility limitation), and high level (imposition of lockdown, closure of schools, workplaces, churches, etc.). The levels of NPIs adherence among vaccinated and unvaccinated individuals were quantified as low (10%), moderate (30%), and high (50%).

## 3. Results

### 3.1. Analytical Results

#### 3.1.1. Computation of Control Reproduction Number

To assess if the implemented control measures, such as vaccination and NPIs, are effective in controlling the COVID-19 outbreak, we computed the control reproduction number, $R_{c}$. The control reproduction number is the average number of COVID-19 secondary infections generated by a single infectious individual when introduced in a mixed population of vaccinated and unvaccinated individuals. We used the next-generation approach as described by Diekmann et al. [40] to compute the control reproduction number of our model.

Let us first define the disease-free equilibrium (DFE) of model (Equation 1) and (Equation 2). At DFE, we have

$E_{u}=I_{p_{u}}=I_{A_{u}}=I_{s_{u}}=C_{v}=R_{u}=D_{u}=E_{v}=I_{p_{v}}=I_{A_{v}}=I_{s_{v}}=C_{v}=R_{v}=D_{v}=0$; $\lambda_{u}=\lambda_{v}=0$; $\dot{S_{u}}>0\;\mathrm{and}\;\dot{S_{v}}>0$.

Hence, the disease-free equilibrium point of our model is given by
$$X_{0}=\left(S_{u}^{0},0,0,0,0,0,0,0,S_{v}^{0},0,0,0,0,0,0,0\right)$$
where $S_{u}^{0}=\frac{\Lambda (\mu +\omega )}{\mu (\mu +\omega +\nu )},\;\mathrm{and}\;S_{v}^{0}=\frac{\Lambda \nu }{\mu (\mu +\omega +\nu )}$.

Let $X={\left(E_{u},I_{p_{u}},I_{A_{u}},I_{s_{u}},C_{u},E_{v},I_{p_{v}},I_{A_{v}},I_{s_{v}},C_{v}\right)}^{T}$ be a vector of infected classes.

Let F be a column vector for all new infections and F=ℑF be the jacobian of F at disease-free equilibrium, $X_{0}$
$$F=\left(\begin{array}{c} S_{u}\lambda_{u}\\ 0\\ 0\\ 0\\ 0\\ S_{v}\lambda_{v}\\ 0\\ 0\\ 0\\ 0 \end{array}\right)\;\;\mathrm{and}\;\;F=\left(\begin{array}{cccccccccc} 0 & A_{1} & A_{2} & A_{3} & A_{4} & 0 & A_{5} & A_{6} & A_{7} & B_{8}\\ 0 & 0 & 0 & 0 & 0 & 0 & 0 & 0 & 0 & 0\\ 0 & 0 & 0 & 0 & 0 & 0 & 0 & 0 & 0 & 0\\ 0 & 0 & 0 & 0 & 0 & 0 & 0 & 0 & 0 & 0\\ 0 & 0 & 0 & 0 & 0 & 0 & 0 & 0 & 0 & 0\\ 0 & B_{1} & B_{2} & B_{3} & B_{4} & 0 & B_{5} & B_{6} & B_{7} & B_{8}\\ 0 & 0 & 0 & 0 & 0 & 0 & 0 & 0 & 0 & 0\\ 0 & 0 & 0 & 0 & 0 & 0 & 0 & 0 & 0 & 0\\ 0 & 0 & 0 & 0 & 0 & 0 & 0 & 0 & 0 & 0\\ 0 & 0 & 0 & 0 & 0 & 0 & 0 & 0 & 0 & 0 \end{array}\right)$$
where
$$\begin{array}{ccc} A_{1}=\frac{S_{u}^{0}b_{uu}\theta_{p_{u}}\left(1-\psi_{u}\right)}{N} & A_{2}=\frac{S_{u}^{0}b_{uu}\theta_{A_{u}}\left(1-\psi_{u}\right)}{N} & A_{3}=\frac{S_{u}^{0}b_{uu}\theta_{s_{u}}\left(1-\psi_{u}\right)}{N}\\ A_{4}=\frac{S_{u}^{0}b_{uu}\theta_{c_{u}}\left(1-\psi_{u}\right)}{N} & A_{5}=\frac{S_{u}^{0}b_{vu}\theta_{pv}\left(1-\psi_{u}\right)}{N} & A_{6}=\frac{S_{u}^{0}b_{vu}\theta_{c_{v}}\left(1-\psi_{u}\right)}{N}\\ A_{7}=\frac{S_{u}^{0}b_{vu}\theta_{s_{v}}\left(1-\psi_{u}\right)}{N} & A_{8}=\frac{S_{u}^{0}b_{vu}\theta_{c_{v}}\left(1-\psi_{u}\right)}{N}\\ B_{1}=\frac{S_{v}^{0}b_{uv}\theta_{P_{v}}\left(1-\psi_{v}\right)}{N} & B_{2}=\frac{S_{v}^{0}b_{uv}\theta_{A_{v}}\left(1-\psi_{v}\right)}{N} & B_{3}=\frac{S_{v}^{0}b_{uv}\theta_{S_{u}}\left(1-\psi_{v}\right)}{N}\\ B_{4}=\frac{S_{v}^{0}b_{uv}\theta_{C_{v}}\left(1-\psi_{v}\right)}{N} & B_{5}=\frac{S_{v}^{0}b_{vv}\theta_{P_{v}}\left(1-\psi_{v}\right)}{N} & B_{6}=\frac{S_{v}^{0}b_{vv}\theta_{A_{v}}\left(1-\psi_{v}\right)}{N}\\ B_{7}=\frac{S_{v}^{0}b_{vv}\theta_{S_{v}}\left(1-\psi_{v}\right)}{N} & B_{8}=\frac{S_{v}^{0}b_{vv}\theta_{C_{v}}\left(1-\psi_{v}\right)}{N} \end{array}$$Let V be the matrix of net transitions
$$V=\left(\begin{array}{c} \left(\alpha_{E}+\nu +\mu \right)E_{u}\\ -\alpha_{E}E_{u}+\left(\alpha_{P}+\mu +\nu \right)I_{P_{u}}\\ -\left(1-\rho_{1}\right)\alpha_{P}I_{P_{u}}+\left(\mu +\nu +\gamma_{a_{1}}+q_{a_{1}}\right)I_{A_{u}}\\ -\rho_{1}\alpha_{P}I_{P_{u}}+\left(\mu +\gamma_{s_{1}}+q_{s_{1}}+\delta_{s_{1}}\right)I_{S_{u}}\\ -q_{a_{1}}I_{A_{u}}-q_{s_{1}}I_{S_{u}}+\left(\delta_{c_{1}}+\gamma_{c_{1}}+\mu \right)C_{u}\\ -\nu E_{u}+\left(\alpha_{E}+\mu \right)E_{v}\\ -\nu I_{P_{u}}-\alpha_{E}E_{v}+\left(\alpha_{P}+\mu \right)I_{P_{v}}\\ -\nu I_{A_{u}}-\left(1-\rho_{2}\right)\alpha_{P}I_{P_{v}}+\left(\mu +\gamma_{a_{2}}+q_{a_{2}}\right)I_{a_{v}}\\ -\rho_{2}\alpha_{P}I_{P_{v}}-\left(\mu +\gamma_{s_{2}}+q_{s_{2}}+\delta_{s_{2}}\right)I_{S_{v}}\\ -q_{a_{2}}I_{A_{v}}-q_{s_{2}}I_{S_{v}}+\left(\delta_{c_{2}}+\gamma_{c_{2}}+\mu \right)C_{v} \end{array}\right)$$The Jacobian matrix (V=ℑV) of matrix *V*, at disease-free equilibrium, $X_{0}$ is given as:
$$V=\left(\begin{array}{cccccccccc} a_{1} & 0 & 0 & 0 & 0 & 0 & 0 & 0 & 0 & 0\\ -\alpha_{E} & a_{2} & 0 & 0 & 0 & 0 & 0 & 0 & 0 & 0\\ 0 & -\alpha_{p}\left(1-\rho_{1}\right) & a_{3} & 0 & 0 & 0 & 0 & 0 & 0 & 0\\ 0 & -\alpha_{p}\rho_{1} & 0 & a_{4} & 0 & 0 & 0 & 0 & 0 & 0\\ 0 & -q_{p_{1}} & -q_{a_{1}} & -q_{s_{1}} & a_{5} & 0 & 0 & 0 & 0 & 0\\ -\nu & 0 & 0 & 0 & 0 & a_{6} & 0 & 0 & 0 & 0\\ 0 & -\nu & 0 & 0 & 0 & -\alpha_{E} & a_{7} & 0 & 0 & 0\\ 0 & 0 & 0 & -\nu & 0 & 0 & -\alpha_{p}\left(1-\rho_{2}\right) & a_{8} & 0 & 0\\ 0 & 0 & 0 & 0 & 0 & 0 & -\alpha_{p}\rho_{2} & 0 & a_{9} & 0\\ 0 & 0 & 0 & 0 & 0 & 0 & -q_{p2} & -q_{a2} & -q_{s2} & a_{0} \end{array}\right)$$
where
$$\begin{array}{ccc} a_{1}=\alpha_{E}+\mu +\nu & a_{2}=\alpha_{p}+\mu +\nu +q_{p1} & a_{3}=\alpha_{p}+\mu +\nu +q_{p1}\\ a_{4}=\gamma_{a1}+\mu +\nu +q_{a1} & a_{5}=\gamma_{c1}+\mu +\delta_{c1} & a_{6}=\alpha_{E}+\mu \\ a_{7}=\alpha_{p}+\mu +q_{p2}\; & a_{8}=\gamma_{a2}+\mu +q_{a2} & a_{9}=\delta_{s2}+\gamma_{s2}+\mu +q_{s2}\\ a_{0}=\gamma_{c2}+\mu +\delta_{c2}. \end{array}$$

Using the next-generation matrix approach, the control reproduction number $R_{c}$ of the model is computed as the spectral radius $\rho \left(F\times V^{-1}\right)$ of the next generation matrix $F\times V^{-1}$, i.e.
$$R_{c}=\rho \left(F\times V^{-1}\right).$$

The expression for the control reproduction number, $R_{c}$, is the sum of three quantities, i.e.,
(4)$$R_{c}=R_{c1}+R_{c2}+R_{c3},$$
where quantities $R_{c1}$ and $R_{c2}$ are contributions of the unvaccinated and vaccinated infectious classes, respectively, while $R_{c3}$ is the contribution from the interaction between the vaccinated and unvaccinated infectious classes to the control reproduction number.

The component $R_{c_{1}}$ is the sum of the four components, i.e., $R_{c_{1Pu}},R_{c_{1Au}},R_{c_{1Su}},\mathrm{and}\;R_{c_{1Cu}}$, which represent the contribution of the pre-asymptomatic, asymptomatic, symptomatic, and confirmed infectious unvaccinated classes, respectively.
(5)$$R_{c_{1}}=R_{c_{1Pu}}+R_{c_{1Au}}+R_{c_{1Su}}+R_{c_{1Cu}},$$
where
$$\begin{array}{cc} R_{c_{1Pu}} & =\frac{\alpha_{E}S_{u}^{0}b_{uu}\left(1-\psi_{u}\right)\theta_{P_{u}}}{2N^{0}a_{1}a_{2}},\\ R_{c_{1Au}} & =\frac{\alpha_{E}S_{u}^{0}b_{uu}\left(1-\psi_{u}\right)\alpha_{p}\left(1-\rho_{1}\right)\theta_{A_{u}}}{2N^{0}a_{1}a_{2}a_{3}},\\ R_{c_{1Su}} & =\frac{\alpha_{E}S_{u}^{0}b_{uu}\left(1-\psi_{u}\right)\alpha_{p}\rho_{1}\theta_{S_{u}}}{2N^{0}a_{1}a_{2}a_{4}},\\ R_{c_{1Cu}} & =\frac{\alpha_{E}S_{u}^{0}b_{uu}\left(1-\psi_{u}\right)\left[a_{3}a_{4}+a_{4}q_{a_{1}}\alpha_{p}\left(1-\rho_{1}\right)+a_{3}q_{s_{1}}\rho_{1}\right]\theta_{C_{u}}}{2N^{0}a_{1}a_{2}a_{3}a_{4}a_{5}}, \end{array}$$$N^{0}$ is the initial population at disease-free equilibrium and is given by $N^{0}=S_{u}^{0}+S_{v}^{0}$.

Similarly, the component $R_{c_{2}}$ is the sum of the four components, i.e., $R_{c_{2Pv}},R_{c_{2Av}},R_{c_{2Sv}},$ and $R_{c_{2Cv}}$, which represent the contribution of the pre-asymptomatic, asymptomatic, symptomatic, and confirmed infectious vaccinated individuals, respectively.
(6)$$R_{c_{2}}=R_{c_{2Pv}}+R_{c_{2Av}}+R_{c_{2Sv}}+R_{c_{2Cv}},$$
$$\begin{array}{cc} R_{c_{2Pv}} & =\frac{\alpha_{E}S_{v}^{0}b_{vv}\left(1-\psi_{v}\right)\theta_{P_{v}}}{2N^{0}a_{6}a_{7}},\\ R_{c_{2Av}} & =\frac{\alpha_{E}S_{v}^{0}b_{vv}\left(1-\psi_{v}\right)\alpha_{p}\left(1-\rho_{2}\right)\theta_{A_{v}}}{2N^{0}a_{6}a_{7}a_{8}},\\ R_{c_{2Sv}} & =\frac{\alpha_{E}S_{v}^{0}b_{vv}\left(1-\psi_{v}\right)\alpha_{p}\rho_{2}\theta_{S_{v}}}{N^{0}a_{6}a_{7}a_{9}},\\ R_{c_{2Cu}} & =\frac{\alpha_{E}S_{v}^{0}b_{vv}\left(1-\psi_{v}\right)\left[a_{8}\left(a_{9}q_{p_{2}}+q_{s_{2}}\alpha_{p}\rho_{2}\right)+a_{9}q_{a_{2}}\alpha_{p}\left(1-\rho_{2}\right)\right]\theta_{C_{v}}}{2N^{0}a_{0}a_{6}a_{7}a_{8}a_{9}}. \end{array}$$

The quantity, $R_{c_{3}}$ is also defined as;
$$R_{c_{3}}=R_{c_{1V}}+\sqrt{{\left[R_{c_{1}}+R_{c_{1V}}-R_{c_{2}}\right]}^{2}+\frac{4}{m_{2}}\left(m_{1}R_{c_{1}}+m_{2}R_{c_{1V}}\right)},$$
where

$R_{c_{1V}}=R_{c_{1Pv}}+R_{c_{1Av}}+R_{c_{1Sv}}+R_{c_{1Cv}}$, $m_{1}=\frac{b_{vu}}{b_{uu}}$, and $m_{2}=\frac{b_{vv}}{b_{uv}}$.

We define $R_{c_{1V}}$ as the sum of four components, i.e., $R_{c_{1Pv}},R_{c_{1Av}},R_{c_{1Sv}},\mathrm{and}\;R_{c_{1Cv}}$, which represent the contribution from the interaction of the unvaccinated individuals with the pre-asymptomatic, symptomatic, and confirmed infectious vaccinated individuals.
$$\begin{array}{cc} R_{c_{1Pv}} & =\frac{\alpha_{E}\nu S_{u}^{0}b_{uv}\left(1-\psi_{v}\right)\theta_{P_{v}}}{2N^{0}a_{1}a_{2}a_{6}a_{7}},\\ R_{c_{1Av}} & =\frac{\alpha_{E}\nu S_{u}^{0}b_{uv}\left(1-\psi_{v}\right)\left[a_{3}\left(1-\rho_{2}\right)\left(a_{2}+a_{6}+a_{6}a_{7}\left(1-\rho_{1}\right)\right)\right]\theta_{A_{v}}}{2N^{0}a_{1}a_{2}a_{3}a_{6}a_{7}a_{8}},\\ R_{c_{1Sv}} & =\frac{\alpha_{E}\nu S_{u}^{0}b_{uv}\left(1-\psi_{v}\right)\alpha_{p}\rho_{2}\left(a_{2}+a_{6}\right)\theta_{S_{v}}}{2N^{0}a_{1}a_{2}a_{6}a_{7}a_{9}},\\ R_{c_{1Cv}} & =\frac{\alpha_{E}\nu S_{u}^{0}b_{uv}\left(1-\psi_{v}\right)\left[a_{3}\left(a_{2}+a_{6}\right)\left(a_{8}\left(a_{9}q_{p_{2}}+\alpha_{P}q_{s_{2}}\rho_{2}\right)+a_{9}\alpha_{P}q_{a_{2}}\left(1-\rho_{2}\right)\right)+D\right]\theta_{C_{v}}}{2N^{0}a_{0}a_{1}a_{2}a_{3}a_{6}a_{7}a_{8}a_{9}} \end{array}$$
where $D=a_{9}a_{6}a_{7}q_{a_{2}}\alpha_{P}\left(1-\rho_{1}\right)$.

#### 3.1.2. Computation of Basic Reproduction Number

In the absence of vaccination and non-pharmaceutical interventions, the control reproduction number reduces to the basic reproduction number, denoted by $R_{0}$, which is given by
$${R_{0}}_{c}=R_{c}|S_{v}^{0}=\nu =\psi_{u}=\psi_{v}=0.$$

The expression for the basic reproduction number, $R_{c_{0}}$, is the sum of two quantities since in the absence of vaccination $S_{v}^{0}=0$, then ${R_{0}}_{c2}=0$.
(7)$${R_{0}}_{c}={R_{0}}_{c1}+{R_{0}}_{c3},$$
where quantities ${R_{0}}_{c1}$ represent the contribution of the unvaccinated infectious classes while ${R_{0}}_{c3}$ is a contribution from the interaction between the vaccinated and unvaccinated infectious classes to the basic reproduction number.

The component, ${R_{0}}_{c1}$ is the sum of the four components, i.e., ${R_{0}}_{c_{1Pu}},{R_{0}}_{c_{1Au}},{R_{0}}_{c_{1Su}}$, and ${R_{0}}_{c_{1Cu}}$, which represent the contribution of the pre-asymptomatic, asymptomatic, symptomatic, and confirmed infectious unvaccinated classes, respectively.
(8)$${R_{0}}_{c1}={R_{0}}_{c_{1Pu}}+{R_{0}}_{c_{1Au}}+{R_{0}}_{c_{1Su}}+{R_{0}}_{c_{1Cu}},$$
where
$$\begin{array}{cc} {R_{0}}_{c_{1Pu}} & =\frac{\alpha_{E}b_{uu}\theta_{Pu}}{2\left(\alpha_{E}+\mu \right)\left(\alpha_{p}+\mu +q_{p1}\right)},\\ {R_{0}}_{c_{1Au}} & =\frac{\alpha_{E}\alpha_{p}b_{uu}\theta_{Au}\left(1-p_{1}\right)}{2\left(\alpha_{E}+\mu \right)\left(\alpha_{p}+\mu +q_{p1}\right)\left(\gamma_{a1}+\mu +q_{a1}\right)},\\ {R_{0}}_{c_{1Su}} & =\frac{\alpha_{E}\alpha_{p}b_{uu}\rho_{1}\theta_{3u}}{2a_{4}\left(\alpha_{E}+\mu \right)\left(\alpha_{p}+\mu +q_{p1}\right)},\\ {R_{0}}_{c_{1Cu}} & =\frac{\alpha_{E}b_{uu}\theta_{Cu}\left(\alpha_{p}q_{a1}\left(1-\rho_{1}\right)a_{4}+q_{p1}\left(\gamma_{a1}+\mu +q_{a1}\right)a_{4}+q_{s1}\rho_{1}\left(\gamma_{a1}+\mu +q_{a1}\right)\right)}{2a_{5}a_{4}\left(\alpha_{E}+\mu \right)\left(\alpha_{P}+\mu +q_{p1}\right)\left(\delta_{c1}+\gamma_{c1}+\mu \right)}. \end{array}$$

The quantity, ${R_{0}}_{c3}$ is also defined as:$${R_{0}}_{c3}=\sqrt{{R_{0}}_{c1}^{2}+\frac{4}{m_{2}}\left(m_{1}{R_{0}}_{c1}\right)}.$$

### 3.2. Numerical Results

Results from the model fitting show a good match between the number of cumulative confirmed cases from the data (red curve) and the number of cumulative confirmed cases obtained from the model (blue curve) for the eight selected countries shown in Figure A1.

#### 3.2.1. Dynamics of COVID-19 Infectious Classes Over Time

We explore how infectious class populations evolve over time with an imperfect vaccine. The estimated model parameters presented in Table A5 and Table A6 were used to compute the ratio of the number of breakthrough infections that would arise among the vaccinated in relation to the unvaccinated subpopulations for the four infectious classes. Figure 2 shows the evolution of the ratio of infected vaccinated individuals to the number of infected unvaccinated for each infectious class (pre-symptomatic, asymptomatic, symptomatic, and confirmed) for the considered countries. Since the COVID-19 vaccines provide partial protection coupled with the emergence of new variants of concern, many breakthrough infections have been reported in most African countries.

Overall, the computed ratios for all eight countries are less than one (varying from 0 to 0.5), indicating that the number of breakthrough infections that arise from the vaccinated infectious classes is relatively lower than that for the unvaccinated. Rwanda and Algeria have the highest ratios, while DR Congo has the least computed ratios (Figure 2).

Results also demonstrated that the ratio of pre-symptomatic and asymptomatic infected individuals increases as the vaccination period increases. For example, in August 2021, the number of vaccinated asymptomatic individuals in Rwanda was 0.28 times that of the unvaccinated. However, this ratio increased to 0.45 by the end of November 2021 (Figure 2). This shows that the proportion of undetected cases (i.e. pre-symptomatic and asymptomatic) for both vaccinated and unvaccinated are generally high in all the African countries as indicated by the blue and red lines (Figure 2).

For all countries, the symptomatic infections among the vaccinated and unvaccinated, except for South Africa, exhibit parallel trends over time. This implies no significant difference in the number of primary infections from unvaccinated and breakthrough infections from the vaccinated symptomatic class, thus demonstrating the vaccine’s effectiveness in reducing the number of both vaccinated and unvaccinated individuals who develop COVID-19 symptoms.

Countries such as Algeria, Namibia, and Libya have similar parallel trends for confirmed and symptomatic cases. The findings from Figure 2 show that DR Congo has the least computed ratios for the four infectious classes due to the low vaccine coverage. For instance, the computed ratio for the number of vaccinated confirmed individuals to unvaccinated is ≈0.0001 as of November 2021. This means that the number of vaccinated individuals confirmed positive for COVID-19 is ≈0. Notably, the number of breakthrough infections from the confirmed vaccinated individuals was higher in countries such as Rwanda, Kenya, South Africa, and Algeria than in other countries.

#### 3.2.2. Impact of Vaccination on the Control Reproduction Number per Country

We computed the values for the control reproduction number, $R_{c}$ using the fixed and estimated model parameters. Overall, the estimates for the $R_{c}$ for the eight African countries during the third wave of the epidemic ranged from 1.911 (for Kenya) to 1.432 (for Libya), with an average of $R_{c}$ = 1.693 (Table A7). Similarly, in the absence of vaccination and other control measures, we computed the values for the basic reproduction numbers, $R_{0}$, for each of the eight countries considered. The results indicated that overall, the values for these countries for $R_{0}$ are approximately two times higher than those for $R_{c}$. With an average of $R_{0}$ = 2.843, estimates for the $R_{0}$ were lowest for DR Congo (2.408) and highest for Algeria (3.640).

#### 3.2.3. Impact of Vaccination on the Transmission Dynamics

We simulated the model to assess the impact of vaccination on the transmission dynamics of COVID-19 in each of the selected eight countries. Results from the simulation (Figure A2) suggest that vaccinated individuals had a much lower force of infection (the rate at which the susceptible individuals become infected per unit time or mass-action transmission) of COVID-19 and a reduced capacity for virus transmission. The reduction in the transmission rate (per capita rate at which two different individuals come in effective contact per unit time) due to vaccination was higher for countries such as Algeria, DR Congo, and Nigeria. However, in Kenya, the force of infection for vaccinated and unvaccinated individuals varies similarly over time.

In addition, we also estimated the infection probabilities and the relative infectiousness to describe the transmission dynamics of vaccinated and unvaccinated populations. The results in (Figure A3) show that the infection probability of disease transmission among the vaccinated population is lower than that among the unvaccinated. Furthermore, the infection probability of disease transmission from infectious unvaccinated to susceptible vaccinated is higher than that from infectious vaccinated to susceptible unvaccinated. The results from Figure A4 also revealed that the pre-symptomatic and asymptomatic vaccinated are less infectious than the vaccinated. However, both the vaccinated and unvaccinated are symptomatic and confirmed to be highly infectious.

#### 3.2.4. Impact of Vaccination on COVID-19 Incidence among the Vaccinated and Unvaccinated Individuals

The vaccine effectiveness (expressed as a detection rate ratio) in reducing new infections for the pre-symptomatic, asymptomatic, and symptomatic cases for the vaccinated as compared to the unvaccinated across the considered African countries are presented in Figure 3a–c. The results from Figure 3 indicate higher heterogeneity in the vaccine efficacy (measured in terms of the detection rate ratio) among the asymptomatic cases compared to the pre-symptomatic and symptomatic cases. The results also showed that higher values for vaccine efficacy against new infections were reported among symptomatic cases (ranging from 0.3 to 5.22) as compared to the asymptomatic (0.118–1.0247) and pre-symptomatic (0.0492–0.89087) cases, indicating lower vaccine efficacy against new symptomatic cases.

The vaccine efficacy against new pre-symptomatic cases was generally less than one across the considered countries. With an expectation of Nigeria, the rest of the countries had detection rate ratio values less than 0.5. In Nigeria, the detection rate of COVID-19 infections among the pre-symptomatic vaccinated was about four times higher than that for the unvaccinated pre-symptomatic cases.

Similarly, the vaccine efficacy against new asymptomatic cases is relatively high (>0.5) for the majority of the considered African countries, as shown in Figure 3b. In DR Congo, there is no significant difference in the detection rates for new infections among vaccinated and unvaccinated individuals. On the other hand, in Algeria, the detection rate among the vaccinated was about 88.2% lower among the vaccinated as compared to the unvaccinated. Results further indicated that countries such as Namibia, South Africa, Libya, and Rwanda had a similar detection rate ratio, implying that vaccinated individuals were at about 25% lower risk of becoming COVID-19 cases compared to the unvaccinated.

The results in Figure 3c also presented the vaccine efficacy against symptomatic cases across the considered African countries. The results indicate that countries with relatively high vaccine coverage, for instance, Rwanda, Kenya, and Algeria, had values for the detection rate ratio greater than one. For instance, in Rwanda, vaccinated people appear to be more than 5-fold to test positive for COVID-19 compared to unvaccinated people. However, in Kenya, since the detection rate ratio is approximately one, vaccinated and unvaccinated individuals are likely to have the same detection rate. On the other hand, in Namibia, the detection rate for the vaccinated symptomatic cases is 65% lower than that for the unvaccinated.

#### 3.2.5. Impact of Vaccination on COVID-19 Mortality among the Vaccinated and Unvaccinated Individuals

The impact of the vaccination on COVID mortality was quantified by the death rate ratio of the vaccinated compared to the unvaccinated for symptomatic and confirmed cases. The results are presented in Figure 3d,e. Our findings indicated a high variation in the vaccine efficacy against COVID-19 deaths among symptomatic and confirmed cases across the considered countries. Overall, vaccination had a higher impact in reducing COVID-19 deaths from confirmed cases compared to those from symptomatic cases, as shown by the lower values of the detection rate ratio for the confirmed. High vaccine efficacy against symptomatic COVID-19 deaths was recorded in the DR Congo (0.0023) and lowest in Rwanda (0.83) and Algeria (0.664). For instance, in DR Congo, the risk of COVID-19 among the symptomatic individuals is 99.76% lower among vaccinated individuals than the unvaccinated. For the confirmed cases, except for DR Congo and Rwanda, the vaccine efficacy against COVID-19 deaths is less than 0.5. A high vaccine efficacy against COVID-19-related deaths from confirmed cases was recorded in Algeria (0.041), South Africa (0.114) and Nigeria (0.132), while low vaccine efficacy was recorded in Rwanda (0.954).

#### 3.2.6. Impact of Vaccine Coverage with Different Levels of Reduction in the Transmission Rate due to NPIs (ψ) among Unvaccinated and Vaccinated Individuals

Figure 4 shows the variation of $R_{c}$ with respect to the vaccine coverage and different levels of reduction in the transmission rate due to adherence to non-pharmaceutical interventions (NPIs) among both vaccinated and unvaccinated individuals for each country. The results obtained, depicted in Figure 4, show that overall, in each of the considered countries, there is a decrease in the values of $R_{c}$ with increasing vaccine coverage combined with a high reduction in the transmission rate due to NPIs by both vaccinated and unvaccinated individuals. The results showed that on average, at least 60% of each African country’s population should be vaccinated to curtail the COVID-19 pandemic (lower the $R_{c}$ below one). Moreover, lower values of $R_{c}$ are possible even when there is a low or moderate reduction in the transmission rate due to NPIs.

For instance, in Algeria, the minimum vaccine coverage required for $R_{c}<1$ is 80%, assuming that there is no reduction in the transmission rate due to NPIs in vaccinated and unvaccinated populations (i.e., ψ=0). However, when the reduction in the transmission rate due to NPIs among the unvaccinated and vaccinated individuals is increased to 10% (low) and 30% (moderate) during the vaccination period, the minimum vaccine coverage required to bring $R_{c}$ to one is reduced to 75% and 70%, respectively.

In DR Congo, findings indicated that at least 65% of the unvaccinated population should be vaccinated for $R_{c}<1$. In addition, the pandemic is curtailed when a 10%, 30%, and 50% reduction in the transmission rate due to NPIs is associated with 58%, 54%, and 50% vaccine coverage. The results further indicated that in most of the countries considered, vaccinating less than 50% (i.e., VC>50%) requires a high reduction in the transmission rate due to NPIs to contain the pandemic. For instance, in Kenya, when the proportion of unvaccinated individuals is below 50%, the $R_{c}>1.5$. To reduce $R_{c}<1$, a high reduction in the transmission rate due to NPIs is required (i.e., ψ>0.75).

## 4. Discussion

In this study, we developed a mathematical model to assess the impact of vaccination programs on curtailing the burden of COVID-19 in eight selected African countries. The model stratifies the total population into two subgroups according to vaccination status. The model is fitted to cumulative daily case data for each selected country corresponding to the third wave of the pandemic. The unknown parameters are estimated using the nonlinear least square method. Overall, the model fits well with the actual cumulative number of confirmed cases for the selected countries.

Our results show the effectiveness of the COVID-19 vaccine against transmission of the SARS-CoV-2 virus to the susceptible contacts from infected vaccinated cases, which is shown by the lower infection probabilities among the vaccinated individuals. These findings are consistent with a study by As et al. [41] that showed a substantial reduction in the transmission risk of PCR-confirmed SARS-CoV-2 infection among vaccinated healthcare workers. Most studies have shown viral load to be an important indicator of the relative infectiousness of both vaccinated and unvaccinated individuals [42]. Our findings suggest that the relative infectiousness of the vaccinated asymptomatic infectious individuals is lower than that of the symptomatic infectious individuals. For example, in South Africa, asymptomatic vaccinated individuals were about three times more infectious than pre-symptomatic. This would be because the asymptomatic individuals shed the virus faster than the pre-symptomatic and symptomatic cases, implying a shorter infectious period [43].

According to the study by Chen et al. [44], asymptomatic cases of COVID-19 are a potential source of substantial spread of the disease, accounting for two-thirds of COVID-19 infections in Africa. Our study findings support this hypothesis, which is indicated by the increased number of both vaccinated and unvaccinated asymptomatic and pre-symptomatic individuals over time. The high proportion of infected individuals without symptoms will likely lead to an under-representation of the number of infections reported in African countries. Therefore, in addition to vaccination, strategies such as mass testing and testing of the asymptomatic close contacts should be implemented to control SARS-CoV-2 transmission in African countries. On the other hand, vaccination programs significantly reduced the number of symptomatic individuals.

The results show that the average control reproduction number across the eight African countries during the first months of vaccination is 1.693, which is greater than one. This suggests that each infectious individual can transmit COVID-19 to two people. The epidemiological implication is that COVID-19 will continue to spread in most African countries even after vaccination but at a slower rate than in the absence of a vaccine. Our findings show that the vaccine’s effectiveness in reducing the detection of new infections is higher among the asymptomatic and pre-symptomatic cases than the symptomatic cases. For instance, in Rwanda, the vaccinated symptomatic individuals are 5-fold more likely to test positive for COVID-19 than the unvaccinated individuals. This implies low vaccine efficacy against new symptomatic infections. This is evidenced by the high detection rates for the symptomatic individuals ranging from 0% to 0.86%. This may be partly due to the low efficacy of the vaccines currently administered to people in many African countries. These findings align with the vaccine impact on new COVID-19 infections reported in clinical trials [45,46].

Vaccination also had a significant impact in reducing COVID-19 deaths that arise from confirmed cases as compared to symptomatic cases. We also observed high variation in the vaccine effectiveness against COVID-19 deaths across all the considered countries. This would be due to different levels of vaccine coverage, economic levels, control measures, testing, and reporting efforts [47]. Furthermore, we observed that countries with high vaccination coverage, such as South Africa and Algeria, had a greater reduction in the mortality rates for confirmed cases compared to largely unvaccinated countries.

Numerical sensitivity analysis performed to evaluate the combined impact of vaccination with different levels of adherence to NPIs showed that to eradicate the pandemic, at least 60% of the population in each African country should be vaccinated, combined with low to high NPI adherence by unvaccinated and vaccinated individuals. Our findings align with the African Centres for Disease Control’s COVID-19 program [48] recommendation that 70% of the population should be fully vaccinated. It should be noted that achieving herd immunity is vital, particularly in the African continent, which is mostly dominated by a large proportion of young people and given the low vaccine supply. However, despite this recommendation and the results of our model, it may take longer for most African countries to reach herd immunity, given the current low vaccine coverage levels in Africa. In addition to the low supply of COVID-19 vaccines [49], many Africans are also unwilling to be vaccinated, as many African countries, for instance, DR Congo, still have thousands to millions of doses that are yet to be administered.

This study has some limitations. We modeled vaccine effectiveness against transmission, new COVID-19 infections, and deaths. However, these estimates may vary according to the significant difference between men and women in the death rate for COVID-19, age structure of the population, specific COVID-19 variants in each country, and multiple COVID-19 vaccine doses administered. In addition, the proposed modeling framework can be extended to include data for all the waves of the pandemic, not just the third wave. Furthermore, the study considered a constant vaccination rate which may not be realistic, as the vaccination rate may depend on the number of vaccine doses available on a particular day. Hence, considering the time-dependent vaccination rate may improve the accuracy of the VE estimates.

## 5. Conclusions

Vaccination programs significantly reduce the transmission as well as relative infectiousness among vaccinated individuals. COVID-19 vaccines prevent COVID-19 disease by reducing the probability of developing symptoms. The study also pointed out that a large proportion of the undetected cases are pre-symptomatic and asymptomatic. This may increase the rate at which susceptible individuals acquire infection, since such individuals are unaware they are sick and are less likely to adhere to NPIs. This study showed that the likelihood of achieving vaccine-derived herd immunity in most African countries is very promising, especially if the vaccination program is complemented with low or moderate levels of adherence to NPIs among both vaccinated and unvaccinated individuals. These results may vary according to the significant difference between men and women in the death rate for COVID-19, the age structure of the population, specific COVID-19 variants in each country, and multiple COVID-19 vaccine doses administered. However, achieving herd immunity in Africa is largely hindered by widespread vaccine hesitancy. It is therefore important for the African governments to design vaccination strategies that address vaccine hesitancy, such as an incentive-based approach, where individuals are given incentives such as food items, beverages, snacks, and T-shirts at the points of vaccination. This may yield positive results for mass vaccination as it will encourage more people to get vaccinated, thus giving indirect protection against the disease to individuals who cannot get vaccinated, such as children, and pregnant women, thus achieving herd immunity.

## Figures and Tables

**Figure 1 vaccines-11-00857-f001:**
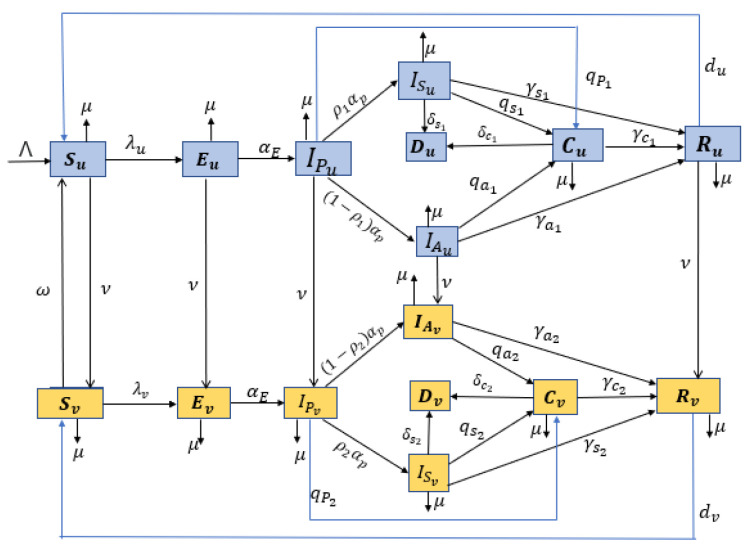
Flowchart of the formulated model.

**Figure 2 vaccines-11-00857-f002:**
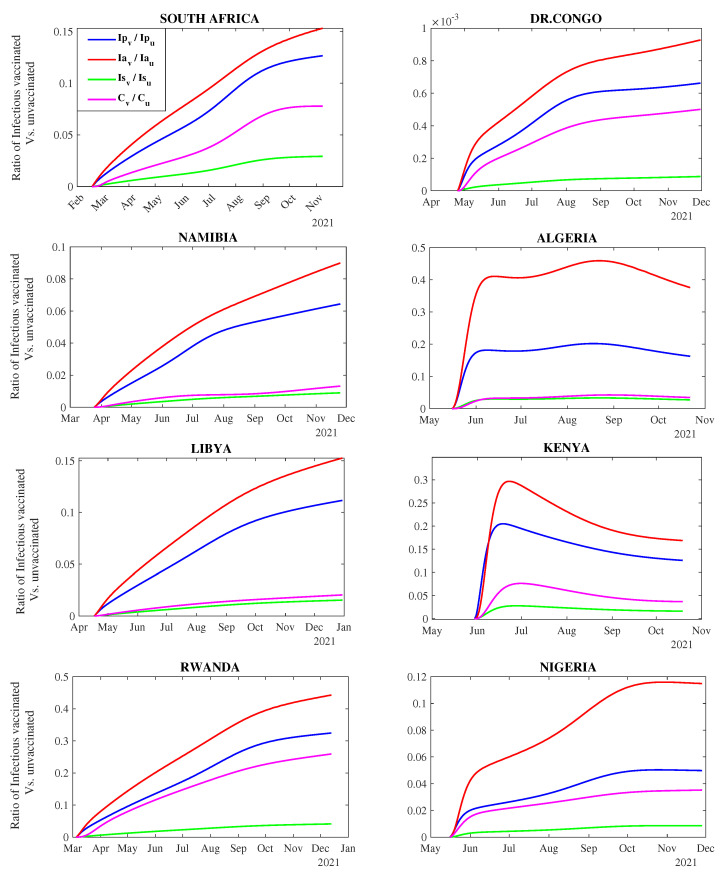
Evolution trend of each infectious compartment for vaccinated and unvaccinated individuals over time in each country. The first line (purple) in the legend depicts the ratio of the number of confirmed infectious vaccinated individuals to confirmed infectious unvaccinated individuals, the second line (green) depicts the ratio of the number of symptomatic infectious vaccinated individuals to the symptomatic infectious unvaccinated, the third line (red) depicts the ratio of the number of asymptomatic infectious vaccinated individuals to asymptomatic infectious unvaccinated, and the last line (blue) depicts the ratio of the number of pre-symptomatic infectious vaccinated individuals to pre-symptomatic infectious unvaccinated.

**Figure 3 vaccines-11-00857-f003:**
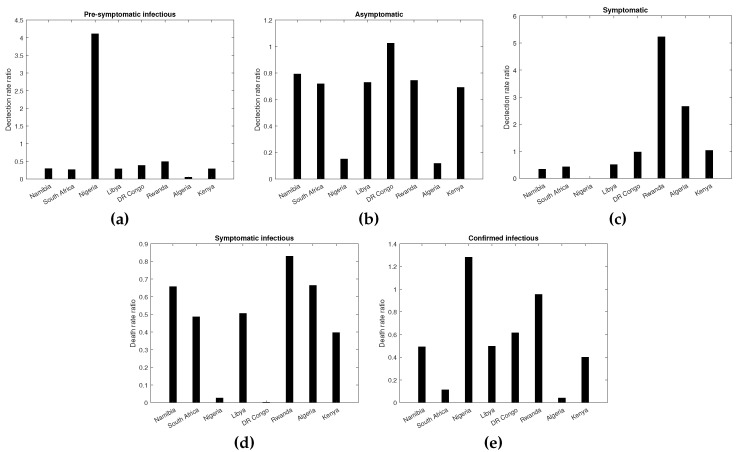
Vaccine effectiveness against COVID-19 infections and deaths. Panels (**a–c**) depict vaccine effectiveness against pre-symptomatic, asymptomatic, and symptomatic new infections Panels (**d,e**) present vaccine effectiveness against symptomatic and confirmed deaths.

**Figure 4 vaccines-11-00857-f004:**
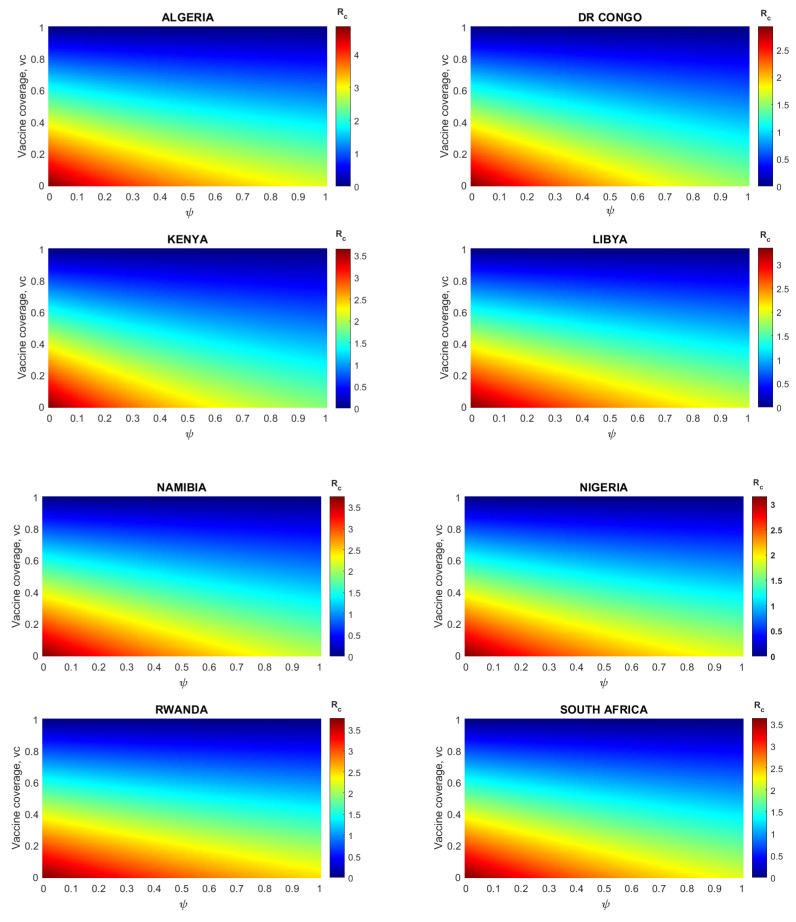
Contour plots of the control reproduction number ($R_{c}$) as functions of the vaccine coverage for different levels of reduction in the SARS-CoV-2 transmission rate due to control measures (ψ) among the unvaccinated and vaccinated individuals.

**Table 1 vaccines-11-00857-t001:** State variables and their description.

State Variable	Description
$S_{u}\left(S_{v}\right)$	Susceptible unvaccinated (vaccinated) population
$E_{u}\left(E_{v}\right)$	Exposed unvaccinated (vaccinated) population
$I_{p_{u}}\left(I_{p_{v}}\right)$	Pre-symptomatic infectious unvaccinated (vaccinated) population
$I_{A_{u}}\left(I_{A_{v}}\right)$	Asymptomatic infectious unvaccinated (vaccinated) population
$I_{S_{u}}\left(I_{S_{v}}\right)$	Symptomatic infectious unvaccinated (vaccinated) population
$C_{u}\left(C_{v}\right)$	Confirmed infectious unvaccinated (vaccinated) population
$R_{u}\left(R_{v}\right)$	Recovered unvaccinated (vaccinated) population
$D_{u}\left(D_{v}\right)$	COVID-deceased unvaccinated (vaccinated) population

**Table 2 vaccines-11-00857-t002:** Description of the fixed and estimated model parameters.

Parameter	Description	Unit
Λ	Recruitment rate	Individual ${\mathrm{day}}^{-1}$
μ	Natural death rate	${\mathrm{day}}^{-1}$
ν	Vaccination rate	${\mathrm{day}}^{-1}$
ω	Vaccine-derived immunity rate	${\mathrm{day}}^{-1}$
$1/\alpha_{E}$	Latent period	days
$1/\alpha_{p}$	Pre-symptomatic period	days
$d_{u}$ ($d_{v}$)	Rate at which recovered unvaccinated (vaccinated) individuals from COVID-19 lose acquired immunity	${\mathrm{day}}^{-1}$
$\rho_{1}\left(\rho_{2}\right)$	Proportion of pre-symptomatic infectious unvaccinated (vaccinated), who develop COVID-19 symptoms	dimensionless
$b_{ij}$	Infection probability of a susceptible individual in class *i* by an infectious individual in class j, for (i,j∈u,v)	dimensionless
$\delta_{s_{1}}\left(\delta_{s_{2}}\right)$	COVID-19 death rate of symptomatic infectious unvaccinated (vaccinated) individuals	${\mathrm{day}}^{-1}$
$\delta_{c_{1}}\left(\delta_{c_{2}}\right)$	COVID-19 death rate of confirmed infectious unvaccinated (vaccinated) individuals	${\mathrm{day}}^{-1}$
$\gamma_{a_{1}}\left(\gamma_{a_{2}}\right)$	Recovery rate of asymptomatic unvaccinated (vaccinated) individuals	${\mathrm{day}}^{-1}$
$\gamma_{s_{1}}\left(\gamma_{s_{2}}\right)$	Recovery rate of symptomatic unvaccinated (vaccinated) individuals	${\mathrm{day}}^{-1}$
$\gamma_{c_{1}}\left(\gamma_{c_{2}}\right)$	Recovery rate of symptomatic unvaccinated (vaccinated) individuals	${\mathrm{day}}^{-1}$
$\theta_{p_{u}}\left(\theta_{A_{u}},\theta_{S_{u}},\theta_{C_{u}}\right)$	Relative infectiousness of unvaccinated pre-symptomatic (asymptomatic, symptomatic, confirmed) individuals	dimensionless
$\theta_{p_{v}}\left(\theta_{A_{v}},\theta_{S_{v}},\theta_{C_{v}}\right)$	Relative infectiousness of unvaccinated pre-symptomatic (asymptomatic, symptomatic, confirmed) individuals	dimensionless
$q_{p_{1}}\left(q_{a_{1}},q_{s_{1}}\right)$	Per capita rate at which unvaccinated individuals from the pre-symptomatic (asymptomatic, symptomatic) infectious class test positive	${\mathrm{day}}^{-1}$
$q_{p_{2}}\left(q_{a_{2}},q_{s_{2}}\right)$	Per capita rate at which vaccinated individuals from the pre-symptomatic (asymptomatic, symptomatic) infectious class test positive	${\mathrm{day}}^{-1}$
$\delta_{s_{1}}\left(\delta_{s_{2}}\right)$	COVID-19 induced death rate of unvaccinated (vaccinated) symptomatic infectious individuals	${\mathrm{day}}^{-1}$
$\delta_{c_{1}}\left(\delta_{c_{2}}\right)$	COVID-19 induced death rate of unvaccinated (vaccinated) confirmed infectious individuals	${\mathrm{day}}^{-1}$

## Data Availability

All the data used are provided within the manuscript.

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
