# Peer review of "Assessing the Impact of Vaccination on the Dynamics of COVID-19 in Africa: A Mathematical Modeling Study"

_vaccines, 2023, doi:10.3390/vaccines11040857_

Round 1

Reviewer 1 Report

The Authors developed a mathematical model to assess the impact of COVID-19 vaccination program to control the burden of OVID-19 in some selected African countries.

The topic is interesting and the resulting considerations are quite intriguing and well reported.

However, some points need to be better explained and some require to be addressed:

1.     Pag 16, line 363: We stimulated the model…, should probabably change to: We simulated….

2.     3.2.3 paragraph, pag 16, in the following sentence: -…vaccinated individuals had much lower rates of the force of infection of COVID-19 and a reduced capacity for virus transmission. The reduction in the transmission rate due to vaccination was higher for countries such as Algeria, DR. Congo, and Nigeria. However, in Kenya, the force of infection for vaccinated and unvaccinated individuals increases at the same rate over time.-

I put in bold some expression used: the Authors should define what force of infection is and if it is different from rate of infection and transmission rate.

3.     In consideration to the significant difference between men and women in the death rate for COVID-19 and also in the severity oif the disease (hospitalizatio, intensive care unit admission) the evaluation of death rate in vaccinated and unvaccinated should beperformed in sex disaggregated manner. With regard to the vaccine coverage, considering the sex bias in access to cure and vaccination in African countries, it is advisable disaggregateby sex the evaluation analysis of vaccine coverage.Thus the developed mathematical model should be revised or adequate for the sex/gender bias.

Author Response

Comment 0: The topic is interesting, and the resulting considerations are quite intriguing and well-reported. However, some points need to be better explained, and some require to be addressed:

Answer: We thank the reviewer for finding the manuscript interesting.

Comment 1: Page 16, line 363: We stimulated the model…, should probably change to: We simulated….

Answer: We thank the reviewer for this interesting comment. The sentence “We stimulated the model…,” on Page 16, line 363 has been changed to: We simulated the model…

Comment 2: 3.2.3 paragraph, page 16, in the following sentence: -…vaccinated individuals had much lower rates of the force of infection of COVID-19 and a reduced capacity for virus transmission. The reduction in the transmission rate due to vaccination was higher for countries such as Algeria, DR. Congo, and Nigeria. However, in Kenya, the force of infection for vaccinated and unvaccinated individuals increases at the same rate over time.-

I put in bold some expressions used: the Authors should define what force of infection is and if it is different from rate of infection and transmission rate.

Answer: We thank the reviewer for this comment. We replied to this comment on page 16, Lines 362-368, as follows: -…vaccinated individuals had a much lower force of infection (the rate at which susceptible individuals become infected by an infectious disease) of COVID-19 and a reduced capacity for virus transmission. The reduction in the transmission rate (number of new infections per unit of time generated by an infected individual) due to vaccination was higher for countries such as Algeria, DR. Congo, and Nigeria. However, in Kenya, the force of infection for vaccinated and unvaccinated individuals varies similarly over time.

Comment 3:  In consideration to the significant difference between men and women in the death rate for COVID-19 and also in the severity of the disease (hospitalization, intensive care unit admission), the evaluation of the death rate in vaccinated and unvaccinated should be performed in a sex-disaggregated manner. With regard to the vaccine coverage, considering the sex bias in access to cure and vaccination in African countries, it is advisable disaggregate by sex the evaluation analysis of vaccine coverage. Thus, the developed mathematical model should be revised or adequate for the sex/gender bias.

Answer: We thank the reviewer for this insightful comment. Indeed, we recognize a significant difference between men and women in the COVID-19 death rate. We designed our work to give an overall trend of COVID-19 patterns in vaccinated and non-vaccinated individuals. With many parameters incorporated into the mathematical model, identifiability problems often arise. That is why we simplify as much as possible the model to consider an overall trend. Moreover, we are now unable to revise the developed mathematical model to consider gender bias because the available data on COVID-19 vaccination and new cases are not disaggregated by sex.

Reviewer 2 Report

Overall, I found the manuscript to be well-written, organized, and informative. The authors developed a mathematical compartmental model to evaluate the impact of vaccination programs on the burden of COVID-19 in eight African countries. They used cumulative case data from the third wave of the pandemic to calibrate the model and performed numerical sensitivity analysis to assess the combined impact of vaccination and control measures on the control reproduction number. The results are insightful and provide important information to policymakers and public health professionals. However, I have a few minor comments that I believe would strengthen the manuscript.

  1. The abstract should briefly summarize the main findings of the study, including the percentage of the population that needs to be vaccinated to eliminate the pandemic and the potential impact of combining vaccination programs with NPI adherence.
  2. The authors state that the vaccine does not reduce the detection of new infections among vaccinated and unvaccinated individuals. This statement is not entirely accurate as the vaccine has been shown to reduce the viral load and the duration of shedding the virus, leading to a lower probability of transmission. The authors should revise this statement or provide additional context.
  3. In the sensitivity analysis section, the authors should clarify how they defined "various levels of NPI adherence" and how they quantified the impact of NPIs on the control reproduction number. Additionally, the authors should briefly discuss the potential challenges and limitations of implementing NPIs in African countries.

Overall, these are minor comments, and I recommend acceptance of the manuscript after the authors address them.

Author Response

Comment 1: The abstract should briefly summarize the main findings of the study, including the percentage of the population that needs to be vaccinated to eliminate the pandemic and the potential impact of combining vaccination programs with NPI adherence.

Answer: We thank the reviewer for this comment. This has been incorporated in the revised manuscript.

Comment 2: The authors state that the vaccine does not reduce the detection of new infections among vaccinated and unvaccinated individuals. This statement is not entirely accurate as the vaccine has been shown to reduce the viral load and the duration of shedding the virus, leading to a lower transmission probability. The authors should revise this statement or provide additional context.

Answer: We thank the reviewer for this comment. This has been addressed. 

Comment 3: In the sensitivity analysis section, the authors should clarify how they defined "various levels of NPI adherence" and how they quantified the impact of NPIs on the control reproduction number. Additionally, the authors should briefly discuss the potential challenges and limitations of implementing NPIs in African countries.

Answer: We thank the reviewer for this comment. The parameter "Psi" is rather a percent reduction in the transmission rate of SARS-CoV-2 due to control measures. We updated the graphs’ axis label and the text in the revised manuscript on pages 17-18, Lines 427-452.

Reviewer 3 Report

1. Re - write your abstract in simple present form

2. Remove unnecessary details from the abstract

3. Check and correct grammatical errors throughout the paper

4. The main motivation of the research is not clear

5. The authors chose some countries from the four regions of Africa, is the choice random? What criteria did they follow in making the choice?

6. The authors divided both vaccinated and un - vaccinated populations in to eight subclasses, why? What is the wisdom behind such classification? How does it help in achieving the desired aim?

7. Are the authors considering two different models with eight compartments each, or a single model with sixteen compartments? Please explain.

8. Enrich the conclusion section by including the limitations of your research.

Author Response

Comment 1: Re-write your abstract in simple present form

Answer: We thank the reviewer for this comment. This has been addressed.

Comment 2: Remove unnecessary details from the abstract

Answer: We thank the reviewer for this comment. This has been addressed.

Comment 3: Check and correct grammatical errors throughout the paper

Answer: We thank the reviewer for this comment. This has been addressed.

Comment 4:  The main motivation of the research is not clear

Answer: We thank the reviewer for this comment. We replied to this comment on page 2, Lines 84-88, as follows: Despite the contributions of these studies, they showed some limitations. Thus, the previous studies ignore the fact that transmission can occur both within vaccinated and unvaccinated people and between vaccinated and unvaccinated people (i.e. an infectious unvaccinated person can infect a vaccinated person and vice versa).

Comment 5: The authors chose some countries from the four regions of Africa, is the choice random? What criteria did they follow in making the choice?

Answer: We thank the reviewer for this comment. We replied to this comment on page 8, Lines 181-185, as follows: Five countries per African region were randomly selected among those for which COVID-19 data are available. However, during the modeling process, two countries, namely Benin and Gabon, were excluded due to the poor quality of the data. Consequently, eight African countries, namely, DR. Congo, Rwanda, Kenya, Algeria, Libya, Namibia, South Africa, and Nigeria, were selected for analysis in this study.

Comment 6: The authors divided both vaccinated and unvaccinated populations into eight subclasses, why? What is the wisdom behind such classification? How does it help in achieving the desired aim?

Answer: We thank the reviewer for this comment. In this study, we hypothesized that COVID-19 vaccination reduces the incidence and mortality of COVID-19. We effectively divided both vaccinated and unvaccinated populations into eight subclasses. This model structure is more realistic than the models used in previous studies. It helps to account satisfactorily for the interaction and transmission between vaccinated and unvaccinated people and between vaccinated and unvaccinated people. It is likely to describe the disease's mechanism and the waning of immunity.

Comment 7: Are the authors considering two different models with eight compartments each or a single model with sixteen compartments? Please explain.

Answer: We thank the reviewer for this comment. We replied to this comment on page 8, Lines 195-198, as follows: a single model with sixteen compartments was used for the calibration. We consider a mixed population where both vaccinated and unvaccinated individuals interact, and the transition from the unvaccinated classes to the vaccinated classes is described by some parameters.

 Comment 8:  Enrich the conclusion section by including the limitations of your research.

Answer: We thank the reviewer for this comment. We replied to this comment on page 20, Lines 536-539.